# AAVAE: Augmentation-Augmented Variational Autoencoders

## Abstract

Recent methods for self-supervised learning can be grouped into two paradigms: contrastive and non-contrastive approaches. Their success can largely be attributed to data augmentation pipelines which generate multiple views of a single input that preserve the underlying semantics. In this work, we introduce augmentation-augmented variational autoencoders (AAVAE), yet another alternative to self-supervised learning, based on autoencoding. We derive AAVAE starting from the conventional variational autoencoder (VAE), by replacing the KL divergence regularization, which is agnostic to the input domain, with data augmentations that explicitly encourage the internal representations to encode domain-specific invariances and equivariances. We empirically evaluate the proposed AAVAE on image classification, similar to how recent contrastive and non-contrastive learning algorithms have been evaluated. Our experiments confirm the effectiveness of data augmentation as a replacement for KL divergence regularization. The AAVAE outperforms the VAE by 30% on CIFAR-10, 40% on STL-10 and 45% on Imagenet. On CIFAR-10 and STL-10, the results for AAVAE are largely comparable to the state-of-the-art algorithms for self-supervised learning.

## 1 Introduction

The goal of self-supervised learning (SSL) (DeSa, 1993) is to learn good representations from unlabeled examples. A good representation is often defined as the one that reflects underlying class structures well. The quality of a representation obtained from SSL is evaluated by measuring downstream classification accuracy on a labelled dataset. In recent years, two families of approaches have emerged as the state-of-the-art for SSL: contrastive and non-contrastive learning.

At its core, a contrastive learning algorithm stochastically creates two views from each training example, called positive and anchor examples, and selects one of the other training examples as a negative (Bachman et al., 2019; Tian et al., 2019; Chen et al., 2020b;a). The positive and anchor examples are brought closer in the representation space, while the negative example is pushed away from the anchor. This definition of contrastive loss brings in two interconnected issues. First, there is no principled way to choose negative examples, and hence these negatives are chosen somewhat arbitrarily each time. Second, the contrastive loss function is not decomposed over training examples because negative examples come from other training examples.

Partly to address these limitations, recent studies have proposed non-contrastive approaches that have removed the need for negative examples (Grill et al., 2020; Caron et al., 2020; Zbontar et al., 2021). These approaches avoid the necessity of explicit negatives by constraining or regularizing dataset-level statistics of internal representation (Zbontar et al., 2021; Caron et al., 2021; Genevay et al., 2018). Dataset-level statistics, which are intractable to compute, are instead approximated using a minibatch of training examples. This often results in the need of large minibatches. Also, the use of batch-level statistics means that non-contrastive losses are not decomposable as well.

Despite the apparent differences between these two families of algorithms, they all recognize the importance of and rely heavily on data augmentation as a way of incorporating domain knowledge. For instance, Chen et al. (2020a) have highlighted that the downstream accuracy after finetuning varied between 2.6% and 69.3% on ImageNet (Deng et al., 2009), depending on the choice of data augmentation. This is perhaps unsurprising since the importance of domain knowledge has been reported in various domains beyond computer vision. In reinforcement learning, Kostrikov et al. (2020) and

Raileanu et al. (2020) have shown the benefit of adding domain information via pixel-level data augmentation in continuous control. In natural language processing, Ng et al. (2020) demonstrate the effectiveness of domain-specific augmentation by using a pretrained denoising autoencoder to build a robust classifier.

A variational autoencoder (VAE) implements a latent variable model using a composition of two neural networks. A neural net decoder maps a latent variable configuration to an observation, and a neural net encoder approximately infers the latent variable configuration given the observation (Kingma & Welling, 2013) . It is often trained to maximize the variational lowerbound or its variant (Kingma & Welling, 2013; Higgins et al., 2016). Careful inspection of this learning objective shows two parts: autoencoding and latent-space regularization. Autoencoding ensures that there is an approximately one-to-one mapping between individual inputs and internal representations. This prevents the collapse of internal representations onto a single point, similar to what negative examples in contrastive learning and regularization of batch-level statistics in non-contrastive learning do. Latent-space regularization, on the other hand, ensures that the internal representation is arranged semantically in a compact subset of the space. It is often done by minimizing the KL divergence (Kullback & Leibler, 1951) from the approximate posterior, returned by the encoder, to the prior distribution and adding noise to the representation during training (i.e., sampling from the approximate posterior). This performs a role similar to that of data augmentation in contrastive and non-contrastive approaches but is different in a way that it is agnostic to the input domain.

Based on these observations: (1) the importance of data augmentations and (2) variational autoencoders for representation learning, we propose a third family of self-supervised learning algorithms in which we augment variational autoencoders with data augmentation. We refer to this family of models as Augmentation-Augmented Variational Autoencoders (AAVAE). In AAVAEs, we replace the usual KL-divergence (Kullback & Leibler, 1951) term in ELBO (Kingma & Welling, 2013) with a denoising criterion (Vincent et al., 2008; 2010) based on domain-specific data augmentation. We hypothesize that this new approach allows the representations learned by AAVAEs to encode domain-specific data invariances and equivariances. The resulting model offers a few advantages over the existing contrastive and non-contrastive methods. First, the loss function is not dependent on the batch-level statistics, which we suspect enables us to use smaller minibatches. Second, the AAVAE does not necessitate an arbitrary choice of negative sampling strategy.

We pretrain AAVAEs on image datasets: CIFAR-10 (Krizhevsky & Hinton, 2009), STL-10 (Coates et al., 2011) and Imagenet (Deng et al., 2009), and as is the norm with other recently proposed approaches (Goyal et al., 2019; Chen et al., 2020a; Caron et al., 2020), we evaluate them on classification tasks corresponding to the dataset using a single linear layer without propagating gradients back to the encoder. We find that our autoencoding-based method gives a downstream classification performance comparable to the current state-of-the-art SSL methods, with $87.14\%$ accuracy on CIFAR-10 and $84.72\%$ on STL-10. On Imagenet, the AAVAE outperforms the carefully crafted pretext tasks for SSL, such as Colorization (Zhang et al., 2016), Jigsaw (Noroozi & Favaro, 2016) and Rotation (Gidaris et al., 2018), demonstrating that designing such complex pretext tasks is unnecessary. As anticipated from our formulation, representation learned by the AAVAE is robust to the choice of hyperparameters, including minibatch size, latent space dimension, and the network architecture of the decoder. Our observations strongly suggest that autoencoding is a viable third family of self-supervised learning approach in addition to contrastive and non-contrastive learning.

## 2 SELF-SUPERVISED LEARNING

Self-supervised learning (SSL) aims to derive training signal from the implicit structure present within data (DeSa, 1993). This enables SSL methods to leverage large unlabeled datasets to learn representations (Goyal et al., 2021) which can then be used to solve downstream tasks, such as classification and segmentation, for which it is often expensive to collect a large number of annotations. Here, we summarize quite a few variations of this approach proposed over the last few years.

### 2.1 PRETEXT TASKS

Pretext tasks are designed to train a neural network to predict a non-trivial but easily applicable transformation applied to the input. For example, Gidaris et al. (2018) randomly rotate an input

image by $0°$, $90°$, $180°$, or $270°$ and train a network to predict the angle of rotation. The colorization pretext task (Zhang et al., 2016) creates a training signal by converting RGB images to grayscale and training a network to restore the removed color channels. Image inpainting (Pathak et al., 2016) learns representations by training an encoder-decoder network to fill in artificially-occluded parts of an image. Both jigsaw (Noroozi & Favaro, 2016) and relative patch prediction (Doersch et al., 2015) tasks divide an input image into patches. The jigsaw task (Noroozi & Favaro, 2016) shuffles the spatial ordering of these patches and trains a network to predict the correct order. In contrast, relative patch prediction (Doersch et al., 2015) selects two patches of an image and asks the network to predict their relative spatial positions. More recently, Doersch & Zisserman (2017) combined various pretext tasks into a single method. Goyal et al. (2019) have, however, shown that training neural network backbones using pretext tasks often does not capture representations invariant to pixel-space perturbations. Consequently, these representations perform poorly on downstream tasks while they solve the original pretext task well.

## 2.2 CONTRASTIVE LEARNING

Between the two major families of state-of-the-art methods for self-supervised learning, we discuss the one based on the so-called contrastive loss function (Hadsell et al., 2006). The contrastive loss is defined such that when minimized, the representations of similar input points are pulled towards each other, while those of dissimilar input points are pushed away from each other. The contrastive loss has its roots in linear discriminant analysis (Fisher, 1936) and is closely related to the triplet loss (Weinberger et al., 2006). Recent approaches in contrastive learning are characterized by the InfoNCE loss proposed by Oord et al. (2018). CPC uses InfoNCE as a lower bound of mutual information (MI) and maximizes this lowerbound, by using negative examples. Deep InfoMax (Hjelm et al., 2018) similarly proposes to use the idea of maximizing MI while considering global and local representations of an image. Hjelm et al. (2018) tested three bounds on MI: Donsker-Varadhan (Donsker & Varadhan, 1983.), Jensen-Shannon (Nowozin et al., 2016), and InfoNCE (Oord et al., 2018), and found that the InfoNCE objective resulted in the best downstream classification accuracies. Since then, several more advances in contrastive self-supervised learning have happened, such as AMDIM (Bachman et al., 2019) and CMC (Tian et al., 2019), both of which focus on using multiple views of each image. Hénaff et al. (2019) extend CPC with an image patch prediction task, and YADIM (Falcon & Cho, 2020) combines these ideas of augmentation and InfoNCE loss from both CPCv2 (Hénaff et al., 2019) and AMDIM (Bachman et al., 2019) under a single framework.

The success of contrastive learning comes from using a large number of negative examples. Misra & van der Maaten (2020) empirically demonstrate with PIRL the benefits of using a large number of negative examples for downstream task performance. PIRL uses a momentum-updated memory bank (Wu et al., 2018) to provide this large cache of negatives. Memory bank models (Wu et al., 2018; Misra & van der Maaten, 2020) need to store and update representations for each data point and hence cannot be scaled up efficiently. To remove the dependence on memory bank, MoCo (He et al., 2020; Chen et al., 2020b) instead introduces a momentum-updated encoder and a comparatively smaller queue of representations to set up positive and negative pairs for contrastive learning. SimCLR (Chen et al., 2020a) removes memory banks and momentum-updated encoders and scales up the batch size to provide a large number of negatives from within each mini-batch. The necessity of a large quantity of negatives for the contrastive loss function to work well proves to be a major challenge in scaling up these methods.

## 2.3 NON-CONTRASTIVE APPROACHES

The second family consists of non-contrastive learning algorithms that aim to learn good representations without negative samples by relying on data-level or batch-level statistics. These algorithms can be classified into two groups: clustering-based (Caron et al., 2018; Asano et al., 2019; Caron et al., 2020) and distillation-based (Grill et al., 2020; Chen & He, 2020; Gidaris et al., 2020; Caron et al., 2021) approaches. A more recently proposed method Barlow Twins (Zbontar et al., 2021) does not fall under either group.

Clustering-based methods, such as DeepCluster (Caron et al., 2018), generate pseudo-labels for training examples by grouping them in the latent space of a neural network. The pseudo-labels are then used to train the neural network. These two steps are repeated several times. Like any

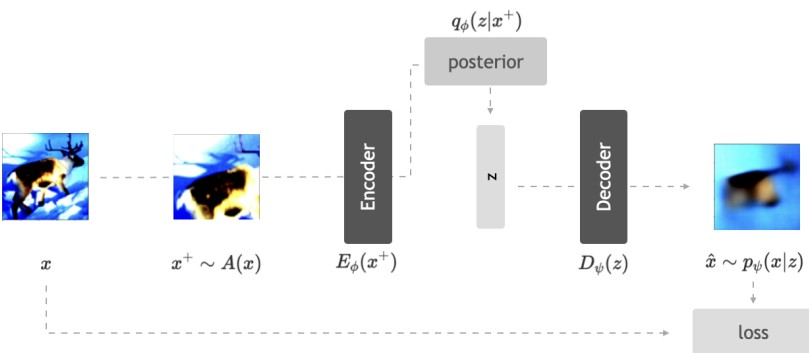

Figure 1: AAVAE: The input to the model is an augmented view of $x^+ \sim A(x)$, the target is the original input $x$. The loss is the reconstruction term of the ELBO (Eq. 3) without the KL-divergence.

classical clustering algorithm, such as $k$-means, this approach exhibits degenerate solutions and requires additional regularization to avoid these solutions. One such degenerate solution is to put all examples into a single cluster. SeLA (Asano et al., 2019) regularizes the clustering process with the Sinkhorn-Knopp algorithm (Cuturi, 2013), encouraging training examples to be equally distributed across the clusters. Caron et al. (2020) extend this approach to use data augmentations and online soft assignments of training examples.

Instead of clustering examples, distillation-based approaches (Tarvainen & Valpola, 2017) rely on having a separate neural network called a teacher network to provide a student network with a target class for each training example. Similar to clustering-based approaches above, this strategy also exhibits trivial solutions, such as the teacher and student networks being constant functions without proper regularization. BYOL (Grill et al., 2020; Richemond et al., 2020; Tian et al., 2020b), and its simpler variant called SimSIAM (Chen & He, 2020), rely on asymmetry in the network architecture between the teacher and student to avoid such degeneracy. To simplify things, SimSIAM (Chen & He, 2020) goes one step further than BYOL (Grill et al., 2020) and removes the momentum-based updates for the teacher network. On the other hand, DINO (Caron et al., 2021) retains the momentum-based updates for the teacher network, replaces the architectural asymmetry with centering of representations of examples within each minibatch, and demonstrates that these techniques combined with a tempered softmax are sufficient regularizers to avoid degeneracy.

Barlow Twins (Zbontar et al., 2021) stands out as an alternative to these two families of approaches. It mixes three principles; (1) batch-level statistics, (2) data augmentation, and (3) whitening (redundancy reduction). At each update, Barlow Twins (Zbontar et al., 2021) normalizes the representations of the training examples within each minibatch to have zero-mean and unit-variance along each dimension. It then tries to maximize the cosine similarity between the representation vectors coming out of a pair of samples drawn from a stochastic data augmentation pipeline applied over a single training example. Finally, Barlow Twins (Zbontar et al., 2021) minimizes the cross-correlation between different coordinates of these vector representations, which amounts to reducing redundancy at the second-order moment. A similar approach has also been proposed by Bardes et al. (2021).

## 3 AUGMENTATION-AUGMENTED VARIATIONAL AUTOENCODERS

Here we revive the idea of autoencoding as a third paradigm for self-supervised learning, in addition to contrastive and non-contrastive learning, which are described in the previous section. In particular, we start from variational autoencoders (VAEs) (Kingma & Welling, 2013) to build a new self-supervised learning algorithm for representation learning. There are three mechanisms by which a VAE captures good representations of data; (1) autoencoding, (2) sampling at the intermediate layer, and (3) minimizing KL divergence (Kullback & Leibler, 1951) from the approximate posterior to the prior distribution, all of which are largely domain-agnostic. We thus introduce domain-specific knowledge by replacing the first mechanism (autoencoding) with denoising (Vincent et al., 2008; 2010) via data augmentation. Furthermore, we remove the third mechanism as we expect KL di-

vergence minimization to be redundant in representation learning. In this section, we explain the original VAE and then carefully describe our proposal of augmentation-augmented VAE.

### 3.1 Training a VAE with the evidence lowerbound (ELBO)

We describe algorithms in this section with the assumption that we are working with images, as has been often done with recent work in self-supervised learning (Oord et al., 2018; Hjelm et al., 2018; Chen et al., 2020a). Hence, let the input $x$ be an image, where $x \in \mathbb{R}^{c \times h \times w}$ with $c$ color channels of height $h$ and width $w$. The VAE then uses a continuous latent variable $z \in \mathbb{R}^d$ to map the high dimensional input distribution, as $p(x) = \int_z p(x|z)p(z)\mathrm{d}z$.

It is however intractable to marginalize $z$ in general, and instead we use a tractable lowerbound to the average log-probability of the training examples. Let $q_\phi(z|x)$ be an approximate posterior distribution to the intractable distribution $p(z|x)$, parametrized by the output of the encoder $E_\phi(x)$. $p_\psi(x|z)$ is a probability distribution over the input $x$, parametrized by the output of the decoder $D_\psi(z)$. The *variational lowerbound* (ELBO) (Kingma & Welling, 2013) to the log-marginal probability $\log p(x)$ is

$$\log p(x) \geq \tilde{\mathcal{L}}(x) = \mathbb{E}_{z \sim q_\phi(z|x)} \left[ \log p_\psi(x|z) + \beta \left( \log p(z) - \log q_\phi(z|x) \right) \right]. \tag{1}$$

The VAE is then trained by minimizing

$$J_{\text{VAE}}(\phi, \psi) = -\frac{1}{N} \sum_{n=1}^{N} \tilde{\mathcal{L}}(x^n), \tag{2}$$

where $x^n$ is the $n$-th training example.

The first term in Eq. 1 serves two purposes. First, it minimizes the reconstruction error, which encourages the intermediate representation of the VAE to be more or less unique for each observation. In other words, it ensures that the internal representations of the inputs do not collapse onto each other. The second purpose, expressed as the expectation over the approximate posterior, is to make the representation space smooth by ensuring a small perturbation to the representation does not alter the decoded observation dramatically.

The second term, the KL divergence (Kullback & Leibler, 1951) from the approximate posterior to the prior, serves a single purpose. It ensures that the representation of any observation under the data distribution is highly likely under the prior distribution. The prior distribution is often constructed to be a standard Normal, implying that the probability mass is highly concentrated near the origin (though not necessarily on the origin). This ensures that the representations from observations are tightly arranged according to their semantics, without relying on any domain knowledge.

### 3.2 Augmentation-augmented variational autoencoder

The AAVAE removes the KL divergence (Kullback & Leibler, 1951) from the formulation because it does not embed domain-specific information and replaces it in favor of an augmented view of the original example. Mathematically, this proposed replacement results in the following loss function:

$$J_{\text{AAVAE}}(\phi, \psi) = \frac{1}{N} \sum_{n=1}^{N} \mathbb{E}_{x_n^+ \sim A(x_n)} \left[ \mathbb{E}_{z \sim q_\phi(z_n|x_n^+)} \left[ \log p_\psi(x_n|z_n) \right] \right], \tag{3}$$

where $A = (a_1, a_2, ..., a_n)$ is a stochastic process that applies a sequence of stochastic input transformations $a_n$. $A$ transforms any input $x$ to generate a view $x^+ \sim A(x)$, while preserving the major semantic characteristics of $x$.

The proposed replacement effectively works by forcing the encoder of the AAVAE to put representations of different views of each example close to each other since the original example must

be reconstructed from all of them. This is unlike the original KL divergence term, which packs the representations globally into the prior. In other words, we replace this global packing with the local packing, where the domain-specific transformations define the local neighborhood. Furthermore, domain-aware transformations have the effect of filling in the gaps between training examples, which indirectly achieves the goal of global packing.

**Comparison to existing approaches**   Compared to the existing approaches, both contrastive and non-contrastive ones, the AAVAE has a unique advantage. AAVAE's loss function is decomposed over the examples, which avoids the need of approximating data-level statistics and computing its gradient for learning. This is advantageous, because we know precisely what we are computing when we use a small minibatch to approximate the gradient of the whole loss function. Generally, this is not the case with algorithms where we need to approximate the gradient of data-level statistics using a small mini-batch. Based on this observation, we expect our approach to be robust to the minibatch size, which we later confirm experimentally in the paper.

A relatively minor but related advantage of the proposed approach over constrastive learning is that there is no need to design a strategy for selecting negatives for each training example. Considering a flurry of recent work reporting on the importance of mining better negative examples (Tian et al., 2020a; Chuang et al., 2020; Robinson et al., 2021), our approach based on autoencoding greatly simplifies self-supervised learning by entirely eliminating negative examples.

## 4   EXPERIMENTS

In this section, we talk about the experiments done for training and evaluating AAVAEs. The design and setup of these experiments are discussed in detail in Appendix A.1.

### 4.1   QUALITY OF REPRESENTATION: DOWNSTREAM CLASSIFICATION ACCURACIES

First, we look at the accuracies from variants of autoencoders, the family to which the proposed AAVAE belongs, presented in the bottom half of Table 1 (left). We consider the vanilla autoencoder (AE), augmention-augmented autoencoder (AAAE), and the variational autoencoder (VAE) as baselines. Our first observation is that there is a significant gap between the proposed AAVAE and all the baselines, with up to 30% points on CIFAR-10, 40% points on STL-10, and 45% points on Imagenet. This demonstrates the importance of data augmentation and noise in the intermediate representation space in making autoencoding a competitive alternative for self-supervised learning. When we add only one of these components, augmentation in the case of AAAEs or sampling in the case of VAEs, we see a big performance degradation from AAVAE. The gap between VAE and AAVAE exposes the inadequacy of KL-divergence as a regularizer for the latent space.

Table 1: Classification performance of Resnet-50 (He et al., 2016) backbone on CIFAR-10 (Krizhevsky & Hinton, 2009), STL-10 (Coates et al., 2011) and Imagenet (Deng et al., 2009) across different methods. All models were pretrained on the corresponding dataset without labels and finetuned using the protocol described in SimCLR (Chen et al., 2020a). The autoencoder trained with our denoising criterion (AAVAE) outperforms the baseline VAE by 30% on CIFAR-10, 40% on STL-10 and 45% on Imagenet. Methods marked with ♣ either use a different backbone than Resnet-50 or a different (non-linear) evaluation strategy.

| Method | CIFAR-10 | STL-10 | Imagenet | Method | Imagenet |
|---|---|---|---|---|---|
| ♣ CPC (large) | - | - | 48.7 | Colorization | 39.6 |
| ♣ CPCv2 | 84.52 | 78.36 | 63.8 | Rotation | 48.9 |
| ♣ AMDIM (small) | 92.10 | 91.50 | 63.5 | Jigsaw | 45.7 |
| ♣ YADIM | 91.30 | 92.15 | 59.19 | BigBiGAN | 56.6 |
| SIMCLR | 94.00 | 92.36 | 69.3 | NPID | 54.0 |
| AE | 56.34 | 42.26 | 0.89 | MoCo | 60.6 |
| AAAE | 50.62 | 41.94 | 1.29 | SwAV | 75.3 |
| VAE | 57.16 | 44.15 | 4.58 | BYOL | 74.3 |
| **AAVAE** | **87.14** | **84.72** | **51.0** | Barlow Twins | 73.2 |

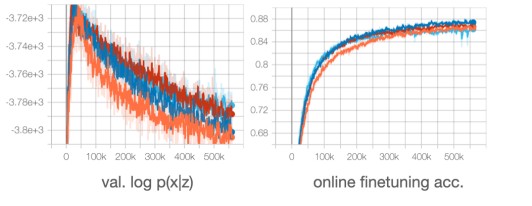 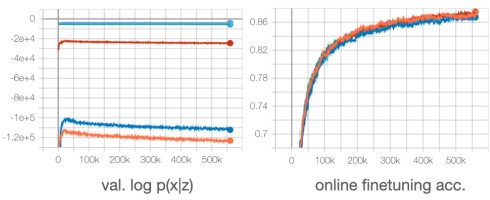

(a) learned distribution width (logscale)  (b) fixed scalar distribution width (logscale)

Figure 2: The AAVAE uses a Gaussian likelihood on pixels for the reconstruction loss with a specified width of the distribution (logscale). In (a), we let the decoder learn the logscale and observe the illusion of overfitting as mentioned in Mattei & Frellsen (2018). In (b), we fix the logscale parameter to an arbitrary scalar by sampling uniformly between [-5, 2]. In both cases, we fail to observe any correlation between the quality of density estimation and learned representation. Plots shown for CIFAR-10 (Krizhevsky & Hinton, 2009) dataset.

We then put the performance of the proposed AAVAE in the context of existing self-supervised learning algorithms presented in the top half of Table 1 (left), and Table 1 (right). We confirm once again what others have observed as to why autoencoding fell out of interest in recent years. All three autoencoder baselines (AE, AAAE, and VAE) severely lag behind the other state-of-the-art self-supervised learning approaches. However, the proposed modification that led to AAVAE significantly narrows this gap on CIFAR-10 and STL-10. On Imagenet, the AAVAE lags behind the current crop of state-of-the-art methods; however, it performs better than any existing pretext task designed for SSL. These results suggest that autoencoding is a viable alternative to contrastive and non-contrastive learning algorithms when designed and equipped appropriately and developed further on from here.

### 4.2 REPRESENTATIONAL QUALITY DOES NOT DETERIORATE

A major downside of the proposed strategy of replacing the KL divergence term in the original loss with data augmentation is that we lose the interpretation of the negative loss as the lowerbound to the log probability of an observation. However, we find it less concerning as the quality of representation is not necessarily equivalent to the quality of density estimation. Furthermore, we make a strong conjecture that the representation quality, which largely depends on the encoder, does not suffer from overfitting (in terms of downstream classification accuracy), even when the quality of density estimation does. Our conjecture comes from the observations that the representation output of the encoder must cope with multiple copies of the same input and noise added in the process of sampling. On the other hand, the decoder can arbitrarily shrink the width of the output distribution per latent configuration, resulting in overfitting to training examples. This conjecture is important since it implies that we should train the AAVAE as long as the computational budget allows, rather than introducing a sophisticated early stopping criterion. More importantly, this would also imply that we do not need to assume the availability of downstream tasks at the time of pretraining.

We test two setups. First, we let the decoder determine the width (in terms of the diagonal covariance of Gaussian) on its own. In this case, we expect the model to overfit the training examples severely, as was observed and argued by Mattei & Frellsen (2018), while the representation quality never deteriorates. In the second setup, we fix the width to an arbitrary but reasonable scalar, which would prevent overfitting in the context of density estimation as long as it is chosen to be reasonably large.

As presented in Fig. 2, in both cases, we observe that the quality of representation, measured in terms of the downstream accuracy, does not deteriorate. Furthermore, as anticipated, we observe that the quality of density estimation quickly overfits in learning the width of output distribution (Figure 2 (a)). Fixing the width to a scalar did not necessarily help avoid the issue of overfitting (Figure 2 (b)). Still, more importantly, we fail to observe any clear relationship between the qualities of density estimation and learned representation.

This finding suggests the need for further study to define and measure the quality of representation distinct from both density estimation quality and downstream accuracy. The former will not only help us measure the learning progress in pretraining time, but will also shed light on what we mean

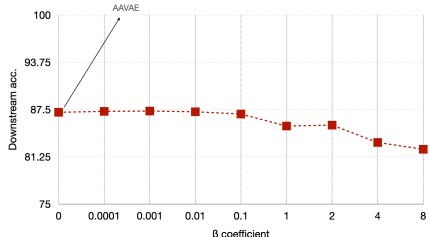

Figure 3: Downstream classification accuracy on CIFAR-10 (Krizhevsky & Hinton, 2009) when we add back KL divergence based regularization with a $\beta$-coefficient (Higgins et al., 2016) to the loss function of AAVAE defined in Eq. 3. We observe a negligible change in the quality of representations, as measured by the classification task, when the KL-term is weighted with a $\beta \ll 1$. For values of $\beta \geq 1$, the quality of representation starts deteriorating, as is seen by the decrease in classification accuracy.

by representation and representation learning. The latter will be needed for future downstream tasks, as the main promise of pretraining is that it results in representations that are useful in the unknown.

### 4.3 COMBINING VAE AND AAVAE

Although we designed AAVAE by *replacing* the KL divergence based regularization with data augmentation based denoising, these two may well be used together. Earlier, Im Im et al. (2017) studied this combination with a simple corruption distribution that is agnostic to the input domain in the context of density estimation. Here, we investigate this combination, with domain-specific transformations, in the context of representation quality.

While keeping the data augmentation based perturbation scheme intact, we vary the coefficient $\beta$ of the KL divergence term. When $\beta = 0$, it is equivalent to the proposed AAVAE. We present the downstream classification accuracies on CIFAR-10 in Figure 3.

We first observe that the KL divergence term has negligible impact when the coefficient is small, i.e., $\beta \ll 1$. However, as $\beta$ grows, we notice a significant drop in the downstream classification accuracy, which we view as a proxy to the representation quality. We attribute this behavior to the tension, or balance, between domain-aware and domain-agnostic regularization of the representation space. As $\beta \to \infty$, the domain-agnostic regularization overtakes and results in the arrangement of the representations that does not reflect the domain-specific structures, leading to worse downstream classification accuracy.

From this experiment, we conclude that for self-supervised pretraining, the proposed approach of data augmentation is a better way to shape the representation space than the domain-agnostic KL divergence based regularization.

### 4.4 HYPERPARAMETER SENSITIVITY

The proposed AAVAE, or even the original VAE, sets itself apart from the recently proposed self-supervised learning methods in that its loss function is decomposed over the training examples (within each minibatch.) Thus, we believe that training the AAVAE is less sensitive to minibatch size, as even with a single-example minibatch, our estimate of the gradient is unbiased. This is often not guaranteed for a loss function that is not decomposed over the training examples. We test this hypothesis by running experiments with varying sizes of minibatches.

As shown in Fig. 4 (a), we observe almost no difference across different minibatch sizes, spanning from 128 to 1024. This is true for both the downstream accuracy (representation quality) and the speed of learning. This is contrary to recent findings from self-supervised learning algorithms, where large minibatches have been identified as an important ingredient (Chen et al., 2020a; Tian et al., 2020b).

This insensitivity to the minibatch size raises a question about other hyperparameters, such as the dimensionality of latent space (Fig. 4 (b)), the decoder architecture (Fig. 4 (c)) and the logscale or

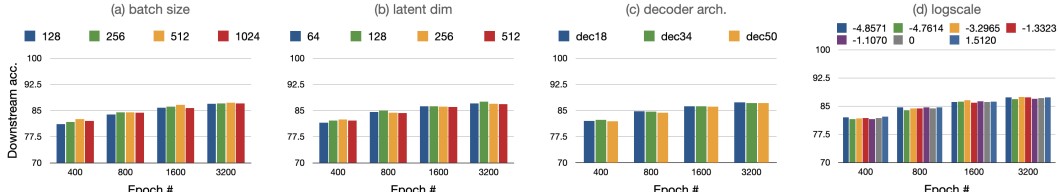

Figure 4: On CIFAR-10 (Krizhevsky & Hinton, 2009), we demonstrate AAVAEs insensitivity to hyperparameters: (a) batch size, (b) latent space dimension, (c) decoder architecture, and (d) logscale parameter (width of the Gaussian likelihood). We vary one specific hyperparameter while keeping the rest fixed for these insensitivity ablations. We select the minibatch size between 128-1024, the dimensionality of the latent space between 64-512, the decoder architecture from decoders that mirror {resnet18, resnet34 or resnet50} encoders, and sample the logscale values from a uniform distribution between [-5, 2].

width of the output distribution (Fig. 4 (d)). We test the sensitivity of the proposed AAVAE to each of these hyperparameters. We find that the quality of representation, measured by the downstream classification accuracy, is largely constant to the change in these hyperparameters. Together with the insensitivity to the minibatch size, this finding further supports our claim that autoencoding-based approaches form a valuable addition to self-supervised learning.

## 5 CONCLUSION

In this paper, we attempt to revive the idea of autoencoding for self-supervised learning of representations. We start by observing that data augmentation is at the core of all recently successful self-supervised learning algorithms, including both contrastive and non-contrastive approaches. We then identify the KL divergence in variational autoencoders (VAE) as a domain-agnostic way of shaping the representation space and hypothesize that this makes it inadequate for representation learning. Based on these two observations: the importance of data augmentations and KL divergence's inadequacy, we propose replacing the KL divergence regularizer with a denoising criterion and domain-specific data augmentations in the VAE and call this variant an augmentation-augmented VAE (AAVAE).

Our experiments reveal that the AAVAE learns substantially better data representation than the original VAE or any other conventional variant, including the vanilla autoencoder and the augmentation-augmented denoising autoencoder. We use downstream classification accuracy from finetuning a linear layer as the metric to measure representation quality and observe more than a 30% improvement on all datasets over the VAE. This result is better than any pretext task for SSL and one of the earlier versions of contrastive learning, CPC. Although the AAVAE still lags behind the more recent methods for SSL, this gap is significantly narrower with the AAVAE than with any other autoencoding variant.

One consequence of autoencoding is that the loss function of AAVAE is decomposed over the examples within each minibatch, unlike contrastive learning (with negative examples from the same minibatch) and non-contrastive learning (which often relies on minibatch statistics). We anticipated that this makes AAVAE learning less sensitive to various hyperparameters, especially the minibatch size. Our experiments reveal that the AAVAE is indeed insensitive to the minibatch size, latent space dimension, and decoder architecture.

Although the proposed AAVAE has failed to outperform or perform comparably to the existing families of self-supervised learning algorithms, our experiments indicate the potential for the third category of self-supervised learning algorithm based on autoencoding. The quality of representations can be significantly pushed beyond that of the vanilla autoencoder and variational autoencoder by making them encode domain specific invariances. Furthermore, autoencoding-based methods, represented by the AAVAE, are robust to the choice of hyperparameters. Based on these observations, we advocate for further research in the direction of autoencoding-based self-supervised learning.

## 6 REPRODUCIBILITY STATEMENT

As mentioned in Section 4.4 of the main paper, our proposed model AAVAE is insensitive to batch size, dimensionality of the latent space, decoder architecture and width of the output distribution (logscale parameter). This in itself makes replicating results for AAVAEs a lot easier across datasets than models that have highly fine-tuned hyperparameters. Then, in Appendix A.1 we list out the encoder/decoder architecture, datasets, augmentation pipeline, optimizer hyperparameters, evaluation protocols etc. that have been used for training AAVAEs and other baseline autoencoder models. Finally, the code used to replicate results from our paper is anonymized and uploaded to the link: https://github.com/aavae-iclr2022/aavae.

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

# A APPENDIX

## A.1 EXPERIMENTAL SETUP

**Architecture** The encoder $\phi$ in our experiments is composed of a residual network backbone (He et al., 2016) followed by a projection layer similar to the one described in (Chen et al., 2020a). The decoder $\psi$ is an inverted version of residual backbone with its batch normalization (Ioffe & Szegedy, 2015) layers removed. We use Resnet-50 as a default option for both the encoder and decoder, but later experiment with varying the decoder architecture.

**Datasets** We test the proposed AAVAE and other more conventional autoencoder models by pre-training them on three datasets: CIFAR-10 (Krizhevsky & Hinton, 2009), STL-10 (Coates et al., 2011) and Imagenet (Deng et al., 2009). CIFAR-10 consists of 50,000 32x32 images in the training set and 10,000 images in the test set. These images are equally divided across 10 labeled classes. For pretraining we use 45,000 image from the training set while 5,000 images are kept for validation. The STL-10 dataset consists of 100,000 unlabelled images resized to 96x96 which are split into 95,000 images for self-supervised pretraining and 5,000 for validation. It further consists of 5,000 training images and 8,000 test images that are labelled across 10 classes. We split the 5,000 training images into 4,500 images for training the downstream classification task and the remaining 500 are kept for validation. Imagenet consists of 1.2 million images in the training split and 50, 000 images in the validation split, spread across 1000 classes. We separate 5000 images from the training set to create our own validation set for finetuning the hyperparameters. The official validation set of Imagenet is what we report the final results on.

**Augmentation pipeline** As mentioned in the paragraph above, we choose image datasets for our experiments with AAVAEs, and hence setup the denoising criterion with an appropriate domain-specific data augmentation pipeline. We define a sequence of common image augmentations $A = \{a_1, a_2, ..., a_n\}$ such as random flip, random channel drop. We also define $a_c$ as a special transform that applies a random resize and crop to an input $x$. Formally, $a_c$ maps $x : \mathbb{R}^{c \times h \times w} \longrightarrow \mathbb{R}^{c \times g \times u}$ where $g \leq h$ and $u \leq w$. For every input $x$ to a AAVAE we define $x^+ \sim A(a_c(x))$ as a view of $x$. The augmentation pipeline defined here is kept the same as that of SimCLR (Chen et al., 2020a), for a fair comparison with other self-supervised learning approaches.

**Optimization and Hyperparameters** We use Adam optimizer (Kingma & Ba, 2014) during pre-training. We use a linear warmup schedule for the learning rate, which is held fixed after the initial warmup. For all our ablation experiments, we keep the weight decay coefficient fixed at 0. When studying the effect of minibatch size, we follow (Goyal et al., 2017) and linearly scale the learning rate and the warmup epoch count with minibatch size. For the hyperparameter sensitivity ablations on CIFAR-10, we vary a particular hyperparameter while keeping the others fixed to their default values. By default, we use a learning rate of $2.5 \times 10^{-4}$, warmup the learning rate until 10 epochs, and keep the batch size at 256. For STL-10 experiments, we set the learning rate at $5 \times 10^{-4}$, warmup epochs count at 10, and keep the batch size at 512. For Imagenet pretraining, we set the total batch size at 512 across 4 GPUs, the learning rate at $5 \times 10^{-4}$, warmup epochs count at 10 and run the pretraining for all autoencoder models until 5 million training iterations.

**Finetuning** Downstream classification accuracy via finetuning has become a widely-used proxy for measuring representation quality. We follow the finetuning protocol put forward by Chen et al. (2020a). After pretraining without any labels, we add and train a linear layer on the pretrained encoder (representation), without updating the encoder. We train the linear layer for 90 epochs with a learning rate defined by: $0.1 * \text{BatchSize}/256$, using SGD with Nesterov momentum.

**Semi-supervised learning evaluation** We run semi-supervised classification task on our models that have been pretrained on the Imagenet dataset. We follow the evaluation process mentioned in previous works (Caron et al., 2020; Zbontar et al., 2021), and train the model on $1\%$ and $10\%$ labeled splits of Imagenet. The training is carried out for 20 epochs with a batch size of 256, using an SGD optimizer with a momentum of 0.9 and no weight decay. Since this is a semi-supervised learning setup with a certain percentage of labels available from the dataset, the backbone is unfrozen during the training process and is trained at a learning rate of 0.01 for the $10\%$ labeled split and at 0.02 for

the 1% labeled split. The linear layer appended on top of the backbone is trained at a learning rate of 0.2 for the 10% labeled split and at a rate of 0.5 for the 1% labeled split.

**Transfer learning tasks**    For the linear classification transfer learning task we use Places205 dataset with the commonly used evaluation protocol (Zbontar et al., 2021; Caron et al., 2020). We train a single linear layer on top of our model for 14 epochs with an SGD optimizer with a learning rate of 0.01, momentum of 0.9 and a weight decay of 5e-4. The learning rate is multiplied by a factor of 0.1 at equally spaced intervals during the training.

For the object detection transfer learning task, we use the VOC07+12 *trainval* set for training and VOC07 test set for eval as previously done by Zbontar et al. (2021). Faster R-CNN with a C4 backbone is used for this downstream task. We train with a batch size of 16 across 8 GPUs for 24000 iterations with a base learning rate of 0.01. We use detectron2 (Wu et al., 2019) library to perform this evaluation.

**Pretraining duration**    As we demonstrate in the paper, the proposed AAVAE benefits from being trained as long as it is feasible. We report the downstream accuracies measured at different points of pretraining. More specifically, we run linear evaluation on our encoder after 400, 800, 1600, and 3200 epochs for the CIFAR-10 experiments. For STL-10, we pretrain our models till 3200 epochs. For Imagenet, we train upto 5 million training steps, which is approximately 2100 epochs.

**Compute and Framework**    All CIFAR-10 (Krizhevsky & Hinton, 2009) experiments are done on a single GPU with a memory size of at least 16GB. All STL-10 experiments are done using two GPUs in the same category. We select GPUs from a mix of NVIDIA RTX 3090s and V100s for CIFAR-10 and STL-10 experiments. Imagenet experiments and downstream evaluations are carried out on 4 A100s. Our codebase uses PyTorch Lightning (Falcon et al., 2019).

## A.2    SEMI-SUPERVISED LEARNING

We finetune the Resnet-50 (He et al., 2016) backbone pretrained by AAVAEs on specified labeled subsets of Imagenet. The two subsets used contain 1% and 10% labeled images of the total number present in the dataset. Table 2 shows the results for the baseline autoencoder models and our proposed AAVAE. The baseline autoencoders are pretty poor in their performance for this semi-supervised evaluation task. In some instances, their performance is 0.1% accuracy on Imagenet, which is equivalent to chance. The AAVAE outperforms the remaining autoencoders considerably on this task with 21.37% accuracy on the 1% labeled subset and a 39.85% accuracy on the 10% labeled subset. However, this is still quite behind when compared against the supervised results or results from other current SSL methods.

## A.3    TRANSFER LEARNING TO OTHER TASKS

For transfer learning to classification tasks, we finetune a linear layer on top of the frozen Resnet-50 backbone pretrained by VAE and AAVAE on Places205 dataset for scene classification. The finetuning protocol is kept the same as the previous works of Zbontar et al. (2021); Misra & van der Maaten

Table 2: Semi-supervised evaluation of Resnet-50 encoder with 1% and 10% labels on Imagenet. Entries with * next to them performed equivalent to chance result on for Imagenet.

|  | Imagenet | |
| --- | --- | --- |
| **Method** | 1% | 10% |
| Supervised | 25.4 | 56.4 |
| SimCLR | 48.3 | 65.6 |
| Barlow Twins | 55.0 | 69.7 |
| BYOL | 53.2 | 68.8 |
| AE | 0.1* | 0.1* |
| AAAE | 0.1* | 0.31 |
| VAE | 0.1* | 0.98 |
| **AAVAE** | **21.37** | **39.85** |

Table 3: Transfer performance of Imagenet pretrained Resnet-50 backbones on classification and object detection tasks. Places205 dataset is used for classification transfer task with the table reporting classification accuracy. For object detection, we use VOC07+12 dataset with Faster R-CNN algorithm and C4 bakcbone.

| Method | Places205 Acc. | VOC07+12 $AP_{all}$ | $AP_{50}$ | $AP_{75}$ |
|---|---|---|---|---|
| Supervised | 51.1 | 53.5 | 81.3 | 58.8 |
| Jigsaw | 41.2 | 48.9 | 75.1 | 52.9 |
| Rotation | 41.4 | 46.3 | 72.5 | 49.3 |
| Barlow Twins | 54.1 | 56.8 | 82.6 | 63.4 |
| VAE | 6.78 | 2.45 | 6.49 | 1.67 |
| **AAVAE** | **41.45** | **15.22** | **35.69** | **10.09** |

(2020). Table 3 shows the results for this downstream evaluation. For comparison, we also include results on Places205 from pretext tasks of Jigsaw (Noroozi & Favaro, 2016) and Rotation (Gidaris et al., 2018), while at the same time including results from one of the current high performers on this evaluation, namely, Barlow Twins (Zbontar et al., 2021).

The finetuning process of object detection transfer task is done on VOC07+12 $trainval$ dataset and the task is evaluated on VOC07 test set. The results are shown in Table 3. Even though the AAVAE performed comparable to the Jigsaw and Rotation pretext tasks on Places205 classification, its performance is greatly affected on the VOC07 detection task. It is far behind the results of these pretext tasks. This result asks whether reconstruction-based SSL techniques are a good fit for transferring representations for object detection tasks. This is something that can be explored in future work.

### A.4 DIRECT INSPECTION OF REPRESENTATION

A major motivation behind our proposal was to use domain-specific data augmentation to encourage representations to encode domain-specific invariances. If AAVAEs indeed reflect such invariances, we expect vector representations coming out of domain-specific perturbations of an individual example to be highly aligned with each other. We test whether this property holds with the AAVAE more strongly than the original VAE by inspecting cosine similarities between pairs of perturbed inputs produced by the same example and between pairs of perturbed inputs produced by different examples. When the former is higher than the latter, we can say the representation encodes domain-specific invariances induced by data augmentation.

In Fig. 5 (a)(i), we make two observations. First, the representation vectors are all extremely aligned for the original VAE. We can interpret this from two perspectives. The first perspective is the so-

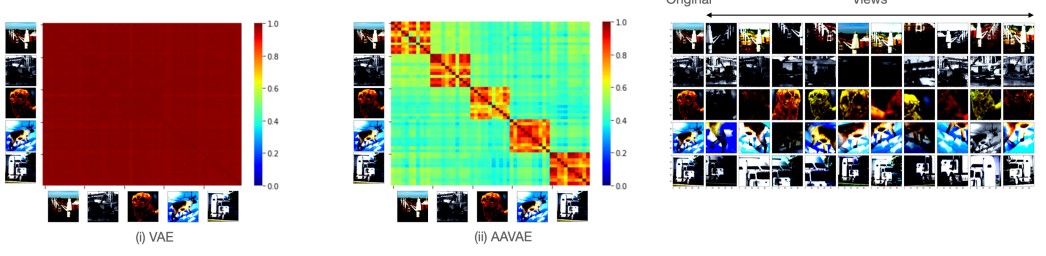

Figure 5: Part (a) shows cosine similarity matrices between pairs of vectors produced by views of the same example and between pairs of vectors produced by views of different examples. We observe a posterior collapse in the case of VAEs in (a)(i). For AAVAEs in (a)(ii), we see strong alignment between views of the same example while the views of different examples are far apart from each other in the representation space. In (b), we show images from the STL-10 dataset (Coates et al., 2011) and their corresponding perturbed versions that generate the cosine similarity matrices in (a).

called posterior collapse (Higgins et al., 2016; Dieng et al., 2019), in which all the approximate posterior distributions, i.e., the representation vectors, are detached from the input and collapse onto each other. The second perspective is the lack of domain-specific invariance, which is evident from the lack of any clusters. Either way, it is obvious that the representations extracted by the original VAE do not reflect the underlying structure of the data well.

On the other hand, with the proposed AAVAE, we see clear patterns of clustering in Fig. 5 (a)(ii). The vectors produced from one example are highly aligned with each other, while the vectors produced from two different examples are less aligned. In other words, the representations capture domain-specific invariances, induced by data augmentation, and the AAVAE does not suffer from posterior collapse. Both these things were well anticipated from the design of our algorithm.

