# OpenReview forum: "AAVAE: Augmentation-Augmented Variational Autoencoders"
_ICLR.cc/2022/Conference — ICLR 2022 Submitted_

### Official Review · Reviewer_cQeC · 2021-10-26

**Correctness:** 4
**Technical Novelty And Significance:** 2
**Empirical Novelty And Significance:** 2
**Recommendation:** 3
**Confidence:** 5

**Main Review:**

Strengths

- The proposed self-supervised method mitigates the requirement for choosing/mining negative examples in a contrastive learning setting and the need for large batch sizes (which is not typically feasible for high dimensional data spaces, e.g., 3D images in medical applications) to derive batch-level statistics for non-contrastive learning methods.
- Data augmentation is a principled approach to incorporate application/task-specific knowledge.
- Comprehsive evaluation against SOTA and ablation experiments within image classification tasks.
- Experiments empircally showcase robustness to hyperparameters including batch size.

Weaknesses

- The significance of decomposing a self-supervised loss over training examples is not well articulated. It is also not clear how the lack of such decomposition negatively impacts contrastive learning methods.
- The paper ignores the large body of work that mines hard negative samples for contrastive learning methods.
- By removing the KL term, the ELBO is not a valid lower bound. Also, the method is not variational anymore.
- The paper lacks novelty as using data augmentation to train an auto-encoder is similar to the denoising criterion for AEs and VAEs.
- The use of data augmentation for self-supervision requires hand-crafted/pre-defined data transformations that are specific to the application at hand, while advantageous to inject domain knowledge, this might lead to learned representations that are not generic enough for multiple downstream tasks.
- Though presented in a different setting, self-supervise learning and injecting domain-specific prior knowledge, the core idea lacks novelty as it is very similar to training auto-encoding based models with a denoising criterion.
- No details provided on the class of transformations used (i.e., A?).
- Convergence and timing for the proposed model is not analyzed in comparison to SOTA, specifically for experiments where it lags or is hardly on par with SOTA.
- Experiments lack evaluation in settings beyond classification, e.g. segmentation, regression.
- While there is a significant improvement compared to AE-based methods, there is a huge performance gap between the proposed method and most of the SOTA considered in the experiments.


**Summary Of The Paper:**

The paper presents a third family for self-supervised learning that relies on generative modeling, in particular variational autoencoders (VAE). The proposed model uses data augmentation in the data space as a way to promote learning richer latent representations, where the augmentation strategy reflects data/domain-specific invariances and equivariances. This augmentation strategy regularizes the VAE  learning process and is used to replace the domain-agnostic KL divergence regularization. Learned representations are evaluated on image classification downstream tasks and compared with recent contrastive and non-contrastive self-supervised methods.

**Summary Of The Review:**

The use of data augmentation and generative model as another family for self-supervise learning has merit.  The proposed methods share the idea of training VAEs with a denoising criterion, albeit removing the KL term in the proposed method. Experiments lack evaluation in settings beyond classification and performance lags behind self-supervised SOTA.

---

> ### Author Response · Authors · 2021-11-17
> **Rebuttal to official review by Reviewer cQeC**
>
> - "The significance of decomposing a self-supervised loss over training examples is not well articulated. It is also not clear how the lack of such decomposition negatively impacts contrastive learning methods."
>
> ----    We can clarify the language in section 4.4 a bit more. By decomposable losses, we mean that the loss is decomposed over elements of the mini-batch (like the reconstruction loss) and the gradient wrt a single sample is not dependent on other samples. If a method is dependent on computing batch-level statistics, then the loss becomes non-decomposable over individual samples in the mini-batch. Decomposable losses result in an unbiased estimate of gradients by each mini-batch, even if it consists of a single element. Non-decomposable loss functions cannot guarantee that the gradient computed over the mini-batch is an unbiased estimate of the gradient for the entire training dataset.
>
> - "The paper ignores the large body of work that mines hard negative samples for contrastive learning methods."
>
> ----    There is an actual principled approach for selecting positive and negative examples for learning, called Boltzmann machine learning. However, it requires us to score every example according to the model and then sample negatives according to this score, making it impractical. We agree with the reviewer that there are other ways of selecting negative samples for self-supervised learning; however, any such process to mine hard negatives cannot avoid this paradigm because the definition of hardness has to be tied together with the model being trained. Selecting negatives this way is often intractable because the dataset size is way too large, so we ultimately have to resort to some arbitrary manual procedure to determine the negatives.
>
> - "By removing the KL term, the ELBO is not a valid lower bound. Also, the method is not variational anymore."
>
> ----    We agree with the reviewer here that after removing the KL-divergence term, it is unclear if we can guarantee that the reconstruction term is still the ELBO. Thus, we did not claim that our proposed method is equivalent to the original VAE and named it differently as Augmentation-Augmented VAE. However, we take into account this suggestion and will consider calling our proposed method as Augmentation-Augmented Denoising Autoencoder. Just for clarification and to help us revise the paper appropriately in the future, would the reviewer tell us what they meant by “variational” here? It is indeed true that we do not optimize an objective over a function space, but that is due to how we parametrize these distributions, including the posterior distribution rather than the choice of the objective function.
>
> - "The use of data augmentation for self-supervision requires hand-crafted/pre-defined data transformations that are specific to the application at hand, … this might lead to learned representations that are not generic enough for multiple downstream tasks."
>
> ----    We agree with the reviewer here but would like to point out that this has been a standard approach to self-supervised learning, regardless of the family of self-supervised learning algorithms, i.e., contrastive or non-contrastive, and now AAVAE. Ultimately we agree with the reviewer that the better idea for us is to learn the correct transformation from data directly.
>
> - "No details provided on the class of transformations used (i.e., A?)."
>
> ----    We refer the reviewer to Appendix A.1 for explanation on the transformations used.
>
> - "Convergence and timing for the proposed model is not analyzed in comparison to SOTA, specifically for experiments where it lags or is hardly on par with SOTA."
>
> ----    We want to thank the reviewer for raising a good point, but it is difficult to compare the convergence of different families of approaches, especially between contrastive methods and reconstruction-based methods. This is something that needs to be done carefully and thoroughly in future works.
>
> - "Experiments lack evaluation in settings beyond classification, e.g. segmentation, regression."
>
> ----    We refer the reviewer to Appendix A.3 for additional downstream task experiments.
>
>
> - "While there is a significant improvement compared to AE-based methods, there is a huge performance gap between the proposed method and most of the SOTA considered in the experiments."
>
> ----    Considering the number of iterative improvements that have been made in other self-supervised learning algorithms vs. autoencoding for representation learning, we still believe that this is the start of a viable direction and asks for more attention from the research community. We believe that this is how innovations happen; we have to try out new ideas even if they are not as mature as some old ones are.

---

### Official Review · Reviewer_Dk62 · 2021-11-01

**Correctness:** 1
**Technical Novelty And Significance:** 1
**Empirical Novelty And Significance:** 1
**Recommendation:** 3
**Confidence:** 5

**Main Review:**

Here are some comments:

- It is not cot clear what authors mean by "batch-level statistics mean that non-contrastive losses are not decomposable as well". Why decomposition is of greater importance?
- Non-contrastive approaches can be trained with relatively smaller batch-sizes, compared to contrastive approaches. See Fig. 3 of BYOL.
- In introduction section, authors motivate the use of VAE as way of getting rid of large batch size and being agnostic to the input domain. However, one paragraph after, they suggest removing the KL and replacing it with a domain-specific loss term for better performance. The design does not follow the motivation.
- While Eq. 1 constitutes a valid ELBO, I'm not sure whether Eq. 3 does. Authors haven't discussed why that results in a valid ELBO.
- I don't agree with the discussion around approximating data-level stats and the requirement of large batch-sizes and how the proposed method handles that.
- While AAVAE shown to be better than other AE methods developed by the authors, there's a huge gap to existing non-contrastive methods, such as BYOL, MoCo, SwAV, and DINO (missing in experiments). I'd argue that I rather prefer 24% better accuracy on ImageNet than a decomposable loss. I don't agree with the argument in Section 4.1; autoencoding is not a viable alternative to non-contrastive methods, while it can be another approach with similar performance to some of pretext tasks designed for SSL.
- Fig. 4(a) suggests that larger batch size does not help significantly. This experiment should be done on ImageNet, where a comparison to other recent SSL method with larger batch-size is possible. In that case, one can conclude with equal batch-size, decomposable loss in AAVAE is beneficial. Also, I believe one can train BYOL with smaller batch-size. For instance, see Fig. 3 of BYOL: Even with batch of 256 it achieves more than 73% accuracy on ImageNet, which is still 22% higher than AAVAE.
- I also encourage authors to go beyond one downstream task of classification. What are the benefit of AAVAE in other tasks such as detection and segmentation?

**Summary Of The Paper:**

This paper proposes a VAE-based approach for the task of self-supervised learning. Authors argue that the need for large batch size can be alleviated by using a augmentation-augmented VAE, which also has the benefit of decomposable loss. In a number of experiments, authors show that AAVAE is comparable to or better than some of the pretext-task based approaches such as Jigsaw and Rotation.

**Summary Of The Review:**

I don't agree that AAVE can be considered as a third family of SSL methods. There is not enough justification around why one would accept 22%-24% lower performance (even with small batch-size) and use AAVE.

---

> ### Author Response · Authors · 2021-11-17
> **Rebuttal to official review by Reviewer Dk62**
>
> - "It is not cot clear what authors mean by "batch-level statistics mean that non-contrastive losses are not decomposable as well". Why decomposition is of greater importance?"
>
> ----    By decomposable losses, we mean that the loss is decomposed over elements of the mini-batch (like the reconstruction loss) and the gradient wrt a single sample is not dependent on other samples. If a method is dependent on computing batch-level statistics, then the loss becomes non-decomposable over individual samples in the mini-batch. Decomposable losses result in an unbiased estimate of gradients by each mini-batch, even if it consists of a single element. With non-decomposable loss functions we often cannot guarantee that the gradient computed over the mini-batch is an unbiased estimate of the gradient for the entire training dataset.
>
> - "Non-contrastive approaches can be trained with relatively smaller batch-sizes, compared to contrastive approaches. See Fig. 3 of BYOL."
>
> ----    We can clarify the language in section 4.4 a bit more, but our intent is not to point out that other methods do not work with small mini-batch sizes. Instead, we want to establish that having a decomposable loss function is advantageous for methods that intend to use a small mini-batch size. For non-decomposable loss functions, the mini-batch estimate of the gradient of the loss function may not be an unbiased estimate of the gradient computed over the entire training set. If the loss function is decomposable, even a single sample in the mini-batch makes the gradient of the loss an unbiased estimate. Thus, with section 4.4, we intend to claim that having a decomposable loss function is advantageous for methods that intend to work with small mini-batches.
>
> - "In introduction section, authors motivate the use of VAE as way of getting rid of large batch size and being agnostic to the input domain. However, one paragraph after, ..."
>
> ----    We would disagree with the reviewer here and argue that the design does follow the motivation. Since VAEs shape the latent space in a domain agnostic fashion, we motivate the need to remove the KL-divergence term and let the data augmentation decide the structure of the latent space, thus injecting domain-specific information into it.
>
> - "While Eq. 1 constitutes a valid ELBO, I'm not sure whether Eq. 3 does. Authors haven't discussed why that results in a valid ELBO."
>
> ----    We agree with the reviewer here that after removing the KL-divergence term, it is unclear if we can guarantee that the reconstruction term is still the ELBO. Thus, we did not claim that our proposed method is equivalent to the original VAE and named it differently as Augmentation-Augmented VAE. However, we take into account this suggestion and will consider calling our proposed method as Augmentation-Augmented Denoising Autoencoder.
>
> - "While AAVAE shown to be better than other AE methods developed by the authors, there's a huge gap to … I rather prefer 24% better accuracy on ImageNet than a decomposable loss."
>
> ----    We disagree with the reviewer on this point. Considering the number of iterative improvements that have been made in other self-supervised learning algorithms vs. autoencoding for representation learning, we still believe that this is the start of a viable direction and asks for more attention from the research community. We believe that this is how innovation happens; we have to try out new ideas even if they are not as mature as some of the old ones are.
>
> - "Fig. 4(a) suggests that larger batch size does not help significantly. This experiment should be done on ImageNet, where a comparison to other recent SSL method with larger batch-size is possible. ... Also, I believe one can train BYOL with smaller batch-size. ..."
>
> ----    We can clarify the language in section 4.4 a bit more, but our intent is not to point out that other methods do not work with small mini-batch sizes. Instead, we want to establish that having a decomposable loss function is advantageous for methods that intend to use a small mini-batch size.  Unfortunately due to limited compute resources we have available, it is not feasible for us to have multiple iterations on ImageNet. We strongly believe our findings on smaller datasets paint valid pictures of the properties of the proposed approach. We expect our labs with greater resources would be able to follow up based on this initial finding to test autoencoder-based approaches on larger scale datasets as well as more diverse set of problems beyond image classification.
>
> - "I also encourage authors to go beyond one downstream task of classification. What are the benefit of AAVAE in other tasks such as detection and segmentation?"
>
> ----    We refer the reviewer to Appendix A.3 for additional downstream task experiments.

---

### Official Review · Reviewer_ebYf · 2021-11-01

**Correctness:** 2
**Technical Novelty And Significance:** 2
**Empirical Novelty And Significance:** 3
**Recommendation:** 3
**Confidence:** 4

**Main Review:**

Postives:
- The method is very simple and practical for using
- The method can be implemented easily
- The performance is very promising
- The hyperparameter study done is very promising

Negatives:
- The paper claims that "In reinforcement learning, Kostrikov et al. (2020) and Raileanu et al. (2020) have shown the benefit of adding domain information via pixel-level data augmentation in continuous control." which is not true. Both those papers simply add common image-level augmentations to the RL pipeline and are not specific to the domain of DM Control suite environment. Similar papers have also used similar methodology [1,2,3] to a variety of different environments (such as procgen) which share minimal domain specificity as DM Control Suite. In fact its probably pretty easy to claim that self-supervised learning works as a domain in-specific algorithm since different works have performed similar methods across a variety of tasks by just changing how the augmentations are added using generic and well-known techniques, where for ex [4] adds simple gaussian noise to state values.
- The paper critiques the self-supervised learning techniques by stating: "First, there is no principled way to choose negative examples, and hence these negatives are chosen somewhat arbitrarily each time." which is misleading. In contrastive learning / triplet learning, there exist many methods that help with mining both the positive and negative samples. See [9] for just a few (of many) different possibilities that may be helpful.
- The paper motivates VAE and the related works for a significant portion of the paper and does not talk about the proposed idea itself.
- By removing the KL loss, is the existent method still a VAE?
- The paper spends alot of time motivating VAEs but ends up removing the KL loss, which is somewhat confusing. Also in Figure 1, I am not sure how the model is updating the posterior if it has gotten rid of the KL loss (and thereby does not have a prior distribution over the latents).
- It would also be useful if the authors can state the differences between a denoising autoencoder (where the noise function is augmentations or A(.)) and AAVAE.
- Further explaining what the paper means by "domain specific transformations" would also be useful. To the reviewers knowledge, augmentations do not need to resemble real life physical transformations, so I am not convinced if augmentations are indeed "domain specific" in any context
- Do the authors have any intuition as to why the VAE outperforms the AAAE since the authors suggest that the reason to drop the KL loss was to make things domain invariant, but its unclear if the KL loss would have helped or not, given that VAE is empirically stronger than AAAE.
- It would be interesting to see Fig 3 with CIFAR-100 instead of CIFAR-10 since CIFAR-10 is possibly to easy of a dataset to require the KL loss. I am not convinced why the KL term + the denoising autoencoder criterion is not helpful since the VAE outperforms the AAAE in the
- The paper misses a few important citations [5-8]



[1] Laskin, Michael, et al. "Reinforcement learning with augmented data." arXiv preprint arXiv:2004.14990 (2020).

[2] https://arxiv.org/abs/2102.11271

[3] http://proceedings.mlr.press/v139/stooke21a.html

[4] https://arxiv.org/abs/2103.06326

[5] http://proceedings.mlr.press/v119/jun20a.html

[6] https://arxiv.org/abs/2105.14859

[7] https://arxiv.org/abs/2010.02014

[8] https://openaccess.thecvf.com/content_CVPR_2020/papers/Zhu_S3VAE_Self-Supervised_Sequential_VAE_for_Representation_Disentanglement_and_Data_Generation_CVPR_2020_paper.pdf

[9] https://github.com/littleredxh/EasyPositiveHardNegative

**Summary Of The Paper:**

The paper presents a new approach to self-supervised learning using VAEs which uses augmentations and a non-contrastive self-supervised objective to perform unsupervised representation learning on the data. The paper compares to state-of-the-art self-supervised methods and underperforms in comparison, but the state-of-the-art methods require significantly more horse power.

**Summary Of The Review:**

The paper currently suffers clarity in explanations. See the main review.

---

> ### Author Response · Authors · 2021-11-17
> **Rebuttal to official review by Reviewer ebYf**
>
> - "The paper claims that "In reinforcement learning, Kostrikov et al. (2020) and Raileanu et al. (2020) have shown the benefit of adding domain information via pixel-level data augmentation in continuous control." which is not true ..."
>
> ----    We clarify here that by domain, we mean the domain of the input, which here is the domain of the input images and not continuous control. Hence we claim that pixel-level data augmentation adds domain-specific information. We can further clarify this point in the revision.
>
>
> - “The paper critiques the self-supervised learning techniques by stating: "First, there is no principled way to choose negative examples, and hence these negatives are chosen somewhat arbitrarily each time." which is misleading."
>
> ----    There is an actual principled approach for selecting positive and negative examples for learning, called Boltzmann machine learning. However, it requires us to score every example according to the model and then sample negatives according to this score, making it impractical. We agree with the reviewer that there are other ways of selecting negative samples for self-supervised learning; however, any such process to mine hard negatives cannot avoid this paradigm because the definition of hardness has to be tied together with the model being trained. Selecting negatives this way is often intractable because the dataset size is way too large, so we ultimately have to resort to some arbitrary manual procedure to determine the negatives.
>
>
> - "By removing the KL loss, is the existent method still a VAE?"
>
> ----    We agree with the reviewer here that after removing the KL-divergence term, it is unclear if we can guarantee that the reconstruction term is still the ELBO. Thus, we did not claim that our proposed method is equivalent to the original VAE and named it differently as Augmentation-Augmented VAE. However, we take into account this suggestion and will consider calling our proposed method as Augmentation-Augmented Denoising Autoencoder.
>
> - "Also in Figure 1, I am not sure how the model is updating the posterior if it has gotten rid of the KL loss (and thereby does not have a prior distribution over the latents)."
>
> ----    The posterior distribution is still updated via the reconstruction error using the reparametrization trick. Although this does not necessarily conform to the standard Normal distribution.
>
>
> - "It would also be useful if the authors can state the differences between a denoising autoencoder"
>
> ----    Our proposed method is similar to the denoising autoencoder with the following differences. First, the type of noise added to the input in our case is structured because it is domain-specific. Second, a denoising autoencoder is deterministic, while our proposed autoencoder is stochastic due to sampling in the latent space.
>
> - "Further explaining what the paper means by "domain specific transformations" would also be useful."
>
> ----    We refer the reviewer to Appendix A.1 for a detailed explanation of domain specific transformations.
>
> - "Do the authors have any intuition as to why the VAE outperforms the AAAE since the authors suggest that the reason to drop the KL loss was to make things domain invariant, but its unclear if the KL loss would have helped or not, given that VAE is empirically stronger than AAAE."
>
> ----    Our hypothesis is that the combination of domain-specific augmentation and a regularized latent space matters. AAAE vs. VAE shows relative importance of one over the other, but it really shows that it is not enough to have only one of these two. Instead, it is critical to have both these components (AAVAE). As we can see in Figure 3 and section 4.3, it is pretty evident that the KL term would not have helped with better representation quality.
>
>
> - "It would be interesting to see Fig 3 with CIFAR-100 instead of CIFAR-10 since CIFAR-10 is possibly to easy of a dataset to require the KL loss."
>
> ----    We thank the reviewer for this suggestion. We have launched the experiments for this and will report back if the experiments are done in time.
>
>
> - "The paper misses a few important citations [5-8]"
>
> ----    We thank the reviewers for pointing out these papers. We are going to go over these papers carefully and add citations at appropriate places in our paper.

---

> > ### Comment · Reviewer_ebYf · 2021-11-23
> > **Thank you for the response**
> >
> > I thank the authors for their time to respond back to my questions and for changing the paper where necessary.
> >
> > However, I remain somewhat unsatisfied with one of the central claims of that paper: adds structured domain specific information, rather than helping with overfitting or performing better function approximation because for overparameterized networks. There have also been recent papers that have tried to do similar things for VAEs (see original review), which is why I will stick with my original review. I believe the paper can be significantly improved, if the paper were to possibly try the same technique with different domains (text, tabular data, etc.) to really show the generality of the method.

---

### Official Review · Reviewer_qnqK · 2021-11-01

**Correctness:** 2
**Technical Novelty And Significance:** 3
**Empirical Novelty And Significance:** 2
**Recommendation:** 5
**Confidence:** 3

**Main Review:**

Strengths:
1. The proposed method is simple to follow.
2. The experiments results show that the proposed method greatly surpasses the existing alternatives.

Weaknesses:
1. By removing the KL in traditional VAE, I think the proposed method can not be called AAVAE since the `V` in VAE has special meanings. It might cause some misunderstanding to the people who are familiar with VAE.
2. Also, when removing KL, I doubt whether AAVAE could sample like VAE dose since I do not find any sampling results in the paper.
3. At last, the features for classification naturally become better when removing KL without considering. Thus, the paper lacks novelty for me.

**Summary Of The Paper:**

This paper introduces a self-supervised learning method augmentation-augmented variational autoencoders (AAVAE) by removing the KL divergence in the traditional VAE. And the experiments on image classification show that the learned features by AAVAE have better properties than the existing alternatives.

**Summary Of The Review:**

See the main review.

---

> ### Author Response · Authors · 2021-11-17
> **Rebuttal to official review by Reviewer qnqK**
>
> - “By removing the KL in traditional VAE, I think the proposed method can not be called AAVAE”
>
> ----    We agree with the reviewer here that after removing the KL-divergence term, it is unclear if we can guarantee that the reconstruction term is still the ELBO. Thus, we did not claim that our proposed method is equivalent to the original VAE and named it differently as Augmentation-Augmented VAE. However, we take into account this suggestion and will consider calling our proposed method as Augmentation-Augmented Denoising Autoencoder.
>
>
> - “Also, when removing KL, I doubt whether AAVAE could sample like VAE dose since I do not find any sampling results in the paper.”
>
> ----     We agree with the reviewer here. The AAVAE does not have a posterior distribution defined by a standard Normal prior, but rather its latent distribution is shaped by data augmentation. This does not allow us to sample from a standard Normal distribution for the latent space. Our goal is to get better representation quality from our model rather than train a better density estimator. However, as seen in Appendix A.4, the AAVAE clusters different views of the same image around a point in the latent space by adding Gaussian noise to the latent representations while training. We can verify this by measuring the cosine similarity between views of an image, as done in Figure 5.

---

### Official Review · Reviewer_61ZD · 2021-11-02

**Correctness:** 3
**Technical Novelty And Significance:** 3
**Empirical Novelty And Significance:** 3
**Recommendation:** 5
**Confidence:** 4

**Main Review:**

- How does your proposed method differ from a denoising autoencoder? Is it fair to say that this method corresponds to a denoising autoencoder where the noise is a random image augmentation and, additionally, noise is added to the latents before reconstructing the image? I find the current name and presentation somehow misleading since by removing the KL divergence, your method is not really optimizing a variational objective anymore, is it?
- By comparing with the AAAE baseline, you tested the importance of your augmentations on the usefulness of the representations. Did you also test what happens if you do not use augmentations only add noise to the latent representations?
- Table 1: What is the difference between the left and right parts of the table? Is the right part an extension of the Imagenet column of the left table?
- 4.2: This section is a bit confusing for me: First I thought you wanted to argue that even though your model suffers from overfitting this is not problematic; then I thought you meant that your model is robust to overfitting, and, finally, it sounded you do the exact opposite with your experiments. If it's the former: I do not agree with your conjecture regarding overfitting. I do not see why the ability of the model to learn the mapping augmented image -> clean image in a robust fashion prevents the model from overfitting. In general, the fact that your model solves some task (and might learn some kind of invariance) to some data points does not allow you to make conclusions about its general ability to solve it. If you mean that your model does not suffer from overfitting I recommend rewriting this paragraph since that's not clear. Regarding your experiments: these results would be more convincing if you trained longer such that the accuracy does not change anymore. Based on this figure one cannot conclude whether the model is prone or robust to overfitting.
- Figure 3: Are these results for a single run or averages over multiple runs? In general, I recommend re-running the experiments - at least for the computationally cheap datasets - for multiple random seeds such that you can report the mean and error of the downstream performance (for all tables and figures, not just Figure 3).
- 4.4. Recent methods like SimSiam also do not need large batch sizes to give reasonably good representations.

**Summary Of The Paper:**

This paper presents a modification to the VAE training scheme. Here, the training objective is modified with the aim of encouraging the model to correctly learn invariances and equivariances of the data. Finally, the authors empirically evaluate the proposed model on standard datasets such as CIFAR-10, STL-10, and ImageNet.

**Summary Of The Review:**

Even though there are some unclarities in the text and experimental results can be improved, the paper contains valuable information for the community. Furthermore, these changes should be feasible to implement during the revision period.

---

> ### Author Response · Authors · 2021-11-17
> **Rebuttal to official review by Reviewer 61ZD**
>
> - “How does your proposed method differ from a denoising autoencoder? Is it fair to say that this method corresponds to a denoising autoencoder where the noise is a random image augmentation and, additionally, noise is added to the latents before reconstructing the image?”
>
> ----    Yes, our proposed method is similar to the denoising autoencoder with the following differences. First, the type of noise added to the input in our case is structured because it is domain-specific. Second, a denoising autoencoder is deterministic, while our proposed autoencoder is stochastic due to sampling in the latent space.
>
>
> - “I find the current name and presentation somehow misleading since by removing the KL divergence, your method is not really optimizing a variational objective anymore, is it?”
>
> ----    We agree with the reviewer here that after removing the KL-divergence term, it is unclear if we can guarantee that the reconstruction term is still the ELBO. Thus, we did not claim that our proposed method is equivalent to the original VAE and named it differently as Augmentation-Augmented VAE. However, we take into account this suggestion and will consider calling our proposed method as Augmentation-Augmented Denoising Autoencoder.
>
>
> - “... Did you also test what happens if you do not use augmentations only add noise to the latent representations?”
>
> ----    We haven't tried adding noise to latent representations without augmentations, but considering the poor performance of autoencoder and augmentation-augmented autoencoder, it is pretty evident how vital sampling in the latent space is. However, the VAE's poor performance shows the importance of data augmentation as well. Based on these two observations, we concluded that it is best to use both of them. That being said, it is an intellectually pressing question raised by the reviewer, and we are going to start experiments on such a variant of the proposed approach. We will report back the results if the experiments finish on time.
>
>
> - “Table 1: What is the difference between the left and right parts of the table? Is the right part an extension of the Imagenet column of the left table?”
>
> ----    The left part of Table 1 reports results for a few methods on three different datasets. These are from the implementations to which we had access and were able to run experiments on additional datasets. The right part of Table 1 reports Imagenet results from the original papers in which they first appeared.
>
>
> - “4.2: This section is a bit confusing for me: First I thought you wanted to argue that even though your model suffers from overfitting this is not problematic; ...”
>
> ----        We can further clarify this section 4.2 in the next revision. There are two types of overfitting in this situation: overfitting in terms of density estimation and overfitting in terms of representation quality, measured using downstream classification accuracy. Our claim here is that these two types of overfitting do not necessarily work together, nor one of them is indicative of the other. Hence, even if the model overfits for density estimation, it doesn't overfit in terms of quality of representation.
>
> - “Figure 3: Are these results for a single run or averages over multiple runs? ...”
>
> ----    These results are posted from a single run, but we can add results from multiple runs for our plots and tables. At the same time, we would like to point out that the results reported in the paper have been observed to be consistent in the experiments conducted over the project's lifecycle, with random values used to initialize the seed, and we are confident that there isn't much variation in these results.
>
>
> - “4.4. Recent methods like SimSiam also do not need large batch sizes to give reasonably good representations.”
>
> ----    We can clarify the language in 4.4 a bit more, but our intent is not to point out that other methods do not work with small mini-batch sizes. Instead, we want to establish that having a decomposable loss function is advantageous for methods that intend to use a small mini-batch size. For non-decomposable loss functions, the mini-batch estimate of the gradient of the loss function may not be an unbiased estimate of the gradient computed over the entire training set. If the loss function is decomposable, even a single sample in the mini-batch makes the gradient of the loss an unbiased estimate. Thus, with section 4.4, we intend to claim that having a decomposable loss function is advantageous for methods that intend to work with small mini-batches.

---

### Decision · Program_Chairs · 2022-01-20

**Decision:**

Reject

**Comment:**

This paper presents an augmentation-based training of autoencoders with stochastic latent space. The proposed method is examined on the representation learning task on several image datasets. While the reviewers found the submission interesting, simple, and easy to implement, they also raised serious concerns around the novelty of the proposed method and the impact of removing the KL term (which removes the generative interpretability of the model). Unfortunately, the experiments do not provide a convincing utility of the model compared to more popular representation learning methods (i.e., contrastive and non-contrastive methods). Given these concerns, the paper is not ready for presentation at ICLR.